# Proteomics of Bacterial and Mouse Extracellular Vesicles Released in the Gastrointestinal Tracts of Nutrient-Stressed Animals Reveals an Interplay Between Microbial Serine Proteases and Mammalian Serine Protease Inhibitors

**DOI:** 10.3390/ijms26094080

**Published:** 2025-04-25

**Authors:** Régis Stentz, Emily Jones, Lejla Gul, Dimitrios Latousakis, Aimee Parker, Arlaine Brion, Andrew J. Goldson, Kathryn Gotts, Simon R. Carding

**Affiliations:** 1Food, Microbiome and Health Research Programme, Quadram Institute Bioscience, Norwich NR4 7UQ, UK; emily.jones@quadram.ac.uk (E.J.); Lejla.Potari-Gul@quadram.ac.uk (L.G.); dimitris.latousakis@quadram.ac.uk (D.L.); aimee.parker@quadram.ac.uk (A.P.); simon.carding@quadram.ac.uk (S.R.C.); 2Department of Metabolism, Digestion and Reproduction, Imperial College London, London SW7 2AZ, UK; 3Core Science Resources, Quadram Institute Bioscience, Norwich NR4 7UQ, UKandrew.goldson@earlham.ac.uk (A.J.G.); kathryn.gotts@quadram.ac.uk (K.G.); 4Norwich Medical School, University East Anglia, Norwich NR4 7TJ, UK

**Keywords:** bacterial extracellular vesicles, proteome, intestine, microbiota, nutrition, *Bacteroides thetaiotaomicron*

## Abstract

Bacterial extracellular vesicles (BEVs) produced by members of the intestinal microbiota can not only contribute to digestion but also mediate microbe–host cell communication via the transfer of functional biomolecules to mammalian host cells. An unresolved question is which host factors and conditions influence BEV cargo and how they impact host cell function. To address this question, we analysed and compared the proteomes of BEVs released by the major human gastrointestinal tract (GIT) symbiont *Bacteroides thetaiotaomicron* (Bt) in vivo in fed versus fasted animals using nano-liquid chromatography with tandem mass spectrometry (LC-MSMS). Among the proteins whose abundance was negatively affected by fasting, nine of ten proteins of the serine protease family, including the regulatory protein dipeptidyl peptidase-4 (DPP-4), were significantly decreased in BEVs produced in the GITs of fasted animals. Strikingly, in extracellular vesicles produced by the intestinal epithelia of the same fasted mice, the proteins with the most increased abundance were serine protease inhibitors (serpins). Together, these findings suggest a dynamic interaction between GI bacteria and the host. Additionally, they indicate a regulatory role for the host in determining the balance between bacterial serine proteases and host serpins exported in bacterial and host extracellular vesicles.

## 1. Introduction

The human gastrointestinal (GI) ecosystem harbours a complex and dynamic population of microorganisms termed the microbiota, which, through its role in digestion, influences host nutrition and energy metabolism, as well as promoting the development and functioning of the immune system [1,2,3]. The GI tract (GIT) microbiota also protects against infection by pathogenic organisms, and it can impact the central nervous system by influencing neural development, neuronal cell signalling, brain chemistry and behaviour, constituting the gut(–microbiota)–brain axis [4,5,6,7,8].

Despite the increasing awareness of the dynamic relationship between the host and its GI microbiota, the molecular basis and pathways of cross-kingdom interactions are poorly understood. In the absence of any direct cognate interactions, they most likely occur via soluble mediators that are able to penetrate the sterile mucus barrier that coats boundary intestinal epithelial cells [9,10]. These include microbial metabolites, signalling molecules, hormones and, from the host, epithelial-derived substances such as mucins, cytokines and antimicrobial peptides [9]. More recently, another pathway of host–microbe crosstalk has been identified involving bacterial extracellular vesicles (BEVs) [11]. These contain not only small molecules such as metabolites [12] but larger molecules including proteins and nucleic acids, with the potential to contribute to inter- as well as cross-kingdom interactions [13,14,15].

BEVs are spherical nanoparticles ranging in size from 20 nm to 400 nm in diameter and include membrane vesicles (MVs), produced by Gram-positive bacteria, and outer membrane vesicles (OMVs), outer–inner membrane vesicles (OIMVs) and explosive outer membrane vesicles (EOMVs) [14,15], produced by Gram-negative bacteria. The GITs of animals contain a multitude of bacterial species capable of producing membrane vesicles, which are implicated in digestion and in the development and functioning of the immune system [15,16]. The Gram-negative anaerobic bacterium *Bacteroides thetaiotaomicron* (Bt) is an abundant bacterial gut symbiont found in the GITs of all animals. In the human distal small intestine and colon, the BEVs that it produces contribute to the degradation of complex polysaccharides and immune homeostasis [17,18,19]. Proteomic studies have shown that members of the *Bacteroides* genus, including Bt, use their BEVs as delivery vehicles for the distribution of hydrolases, such as proteases and glycosidases [20]. More recently, we have shown that Bt BEVs produced in the GI tracts of mice are enriched in proteins and enzymes, including dipeptidyl peptidases, asparaginase and bile salt hydrolases, which can influence host cell biosynthetic pathways [21].

EVs are released by host cells in the intestinal mucosa, including immune and epithelial cells [22], and can be exogenously derived when associated with ingested food [23]. Three main categories of human cell-derived EVs have been distinguished based on their size: exosomes (30–150 nm), microvesicles (100–1000 nm) and apoptotic bodies (1000–5000 nm). Exosome production involves plasma membrane endocytosis, multivesicular endosome trafficking and exocytosis into the extracellular milieu. Microvesicles are produced through plasma membrane budding, while apoptotic bodies are derived from membrane blebbing during apoptosis [23]. Host intestinal EVs can theoretically contain all molecules present in the cell of origin, including lipids and sphingolipids, proteins, mRNAs, microRNAs and non-coding RNAs [24]. EVs can protect their cargo from proteases and nucleases, which enables the delivery of bioactive molecules to neighbouring cells in the GIT, enabling them to act as mediators of long-distance cell–cell communication and interaction [25].

Among their protein cargo, EVs can contain serpins (serine protease inhibitors), a family of proteins that inhibit serine proteases by irreversibly changing their conformation [26]. Mammalian serpins are involved in diverse physiological processes, including blood coagulation, blood pressure, fibrinolysis, insulin sensitivity, inflammatory or immune responses and tissue remodelling [26,27]. Several serpins were identified in EVs in a study assessing their ability to promote wound closure in a mouse model of type 2 diabetes (db/db) with impaired wound healing. While wild-type EVs collected at the wound site of a WT mouse accelerated wound closure in a db/db mouse, EVs from diabetic mice were impaired in their ability to heal wounds. A comparative proteomics study showed a deficit in the abundance of a family of serine protease inhibitors, specifically serpin A1 (anti-trypsin), serpin F2 (anti-plasmin) and serpin G1 (plasma protease C1 inhibitor), in EVs from diabetic mice [24]. Another example is the serpin family E member 1 (SERPINE1), enriched in EVs released by senescent tumour cells (STCs), in the context of EVs emerging as mediators of the senescence-associated secretory phenotype (SASP) tumour-promoting effect. Through its packaging in EVs, SERPINE1 is transported into recipient cancer cells, binds to NF-κB p65 and promotes its nuclear translocation, which results in tumour progression [28].

Here, we assessed the impact of nutrient stress on the BEV protein composition by performing a comparative proteomics analysis of BEVs produced in the intestines of Bt mono-conventionalised germ-free mice provided food ad libitum or fasted. Our results establish that the abundance of serine proteases, including dipeptidyl peptidase-4 (DPP-4), is decreased under fasting conditions. These changes were mirrored by the increased abundance of serine protease inhibitors (serpins) in host cell-derived EVs.

## 2. Results

### 2.1. The Proteome of BEVs Produced In Vivo

To assess whether nutrient stress and a period of fasting affect the protein composition of BEVs produced by Bt in the GIT, germ-free mice were orally gavaged with Bt, with one group of conventionalised mice allowed unrestricted access to food and water and a second group with restricted access to food for 16 h (the maximum period allowed under UK Home Office Regulations). It should be noted that the vesicles extracted from the caecum under these conditions represent a mixture of BEVs and host-derived EVs, as the current isolation methods do not fully separate these populations. BEVs and EVs extracted from the caecum were equivalent in size from both fed and fasted mice, with a mean size of approximately 195 (±9.5) nm as determined by nanoparticle tracking analysis (NTA) (Figure 1a). By contrast, 1.6 times more nanoparticles were recovered from the caeca of fasted mice (1.0 × 10^11^ particles/mL) compared to fed animals (Figure 1a). Vesicles obtained from both conditions were visualised by electron microscopy (Figure 1b).

A comparison of the proteomes of the caecal BEVs from fed and fasted animals (Appendix A) showed differences in protein abundance (Figure 1c). Using principal component analysis (PCA), the first two PCs accounted for 68.6% and 14.5%, respectively, of the total variation in the dataset, allowing a clear distinction between the BEV proteomes from fasted and fed mice (Figure 1c). A volcano plot analysis of the dataset showed that, of the 528 proteins identified in BEVs extracted from fasted and fed mice, 263 were differentially abundant (fold change ≥ 1.3, *p* ≤ 0.05). Of these, 119 were more abundant in BEVs extracted from fasting animals, whereas 144 proteins were more abundant in BEVs derived from fed animals (Figure 1d).

An overrepresentation analysis was performed to determine whether known biological functions or processes were enriched in BEVs compared to their parental cell’s proteome (Appendix A). In general, activities related to the hydrolysis of sugar molecules were overrepresented in BEVs. For example, sucrose alpha-glucosidase activity (GO:0004575) and beta-fructofuranosidase activity (GO:0004564) were enriched in BEVs (fold change 10.46 for both annotations). Peptidase activities were also enriched, with dipeptidyl peptidase activities enriched by a factor of 7.47 in BEVs (Appendix A).

Many proteins encoded by polysaccharide utilisation units (PULs) comprising neighbouring genes involved in the breakdown of specific glycans were present in BEVs (Appendix A), which were classified using the Polysaccharide-Utilisation Loci DataBase (PULDB) http://www.cazy.org/PULDB/ (accessed on 21 April 2025) [29] (Appendix A). The proteins originating from the starch-degrading PUL66 were particularly abundant (peptide–spectrum match or PSM ranging from 5 to 41), which most likely reflects the high (~34%) starch content of mouse chow. The PUL components involved in the degradation of rhamnogalacturonan-II (PUL77), pectic galactan (PUL86) and arabinogalactan (PUL65) were also found to be abundant. Moreover, in fasted mice, there was an increased abundance of PULs capable of degrading host glycans and mucins (PULs 6, 35, 17 and 81).

On a different note, Bt-TenA (BT_3146), which belongs to a novel sub-class of TenA_C enzymes and includes members in animal genomes [30], showed a 2.2-fold increase in abundance in BEVs from fasted mice. As Bt-TenA contributes to the thiamine salvage pathway, which regenerates thiamine-derived pyrimidine from degraded thiamine products [31], the increased abundance of Bt-TenA may ensure the survival of Bt under nutrient-poor conditions and reduced thiamine bioavailability.

### 2.2. Serine Proteases Are Less Abundant in BEVs Produced in Fasted Mice

We next performed a gene ontology analysis and pathway enrichment to investigate whether certain categories of proteins were affected by the diet within the proteomes of the BEVs under each condition. Strikingly, the abundance of nine of ten proteins identified as serine proteases was reduced in BEVs produced in the intestines of fasted mice (Table 1). Amongst them were DPP-4-like proteins, predicted to be secreted and encoded by BT_4193 and BT_3254. In humans, DPP-4 (CD26) truncates proteins containing the amino acids proline or alanine in the second position of the N terminus, including the appetite hormones GLP-1, GIP and neuropeptide substance P. DPP-4-like activity encoded by the intestinal microbiome may therefore constitute a novel mechanism to modulate protein digestion and host metabolism [32]. To date, no defined roles have been attributed to the eight remaining serine proteases.

### 2.3. BT_1314 Is a γ-D-Glutamyl-L-Diamino Acid Endopeptidase

We have previously shown the BT_1314 enzyme (annotated as a dipeptidyl-peptidase-6) to be enriched in BEVs in vivo compared to BEVs harvested from in vitro cultures [21]. Similarly to DPP-4 (BT_4193) and the prolyl tripeptidyl aminopeptidase BT_3254 [33], BT_1314 decreased in abundance (0.838-fold) in BEVs produced in fasted mice (Appendix A). The human DPP-6 (DPP-4-like) is a serine protease that binds specific voltage-gated potassium channels, altering their expression and biophysical properties [34], whereas BT_1314 is not a serine protease (as defined by the PANTHER classification system) and is therefore misannotated. A BLAST (version 2.14.0+) of the BT_1314 protein sequence against the “UniProtKB with 3D structure (PDB)” database revealed 29.35% identity to the protein sequence of YkfC from *Bacillus cereus* ATCC 10987 [35], with a shared protein domain architecture that includes an SH3_16 domain (bacterial dipeptidyl-peptidase SH3 domain) and a C-terminal NLPC_P60 endopeptidase domain, as predicted by InterPro [36] (Figure 2a). Using a recombinant version of BT_1314, we confirmed that BT_1314 is, like YkfC of *Bacillus* sp., a γ-D-glutamyl-L-diamino acid endopeptidase that is able to degrade L-Ala-γ-D-Glu-diaminolimelic acid (Tri-DAP) (Figure 2b), a substrate derived from the peptidoglycan (PG) of a subset of Gram-negative bacilli and Gram-positive bacteria including *Bacillus*, which is an agonist of the NOD1 pattern recognition receptor [37].

### 2.4. The Serine Protease BT_4193 Is a Functional DPP-4 Enzyme in BEVs

Amongst the potential DPP-4 enzymes selectively enriched in BEVs produced in vivo in the mouse GIT [21], we have established that BT_1314 is misannotated as DPP-6- or DPP-4-like (it does not degrade the DPP-4 substrate Ala-Pro-pNA) and is instead a murein peptide peptidase. In addition, Keller et al. have demonstrated that BT_3254 is a prolyl tripeptidyl aminopeptidase (PTP) and that BT_4193 is a DPP-4 and a functional homologue of the human DPP-4 that can be inhibited by human DPP-4-targeted drugs [33]. A non-catalytic role for BT_4193 was also described in maintaining cell envelope integrity and establishing Bt within multispecies communities [33].

To demonstrate that BEVs produced in vivo can actively degrade both DPP-4 and PTP substrates, chow-fed germ-free mice were mono-conventionalised with either wild-type Bt, a DPP-4 deletion mutant or a PTP deletion mutant. To overcome any interfering activity of the mouse EVs with BEV DPP-4 activity, we used sitagliptin, a DPP-4 inhibitor that is active against mammalian DPP-4 but with little effect on microbial DPP-4 [33,38]. Our results confirmed that BEV-derived BT_4193 accounted for DPP-4 activity, with the BT_4193 mutant showing a significant decrease in activity (*p* = 0.0154) and with the activity being further decreased in the presence of sitagliptin (Figure 3a). Some residual activity was detectable for DPP-4-mutant-derived BEVs when compared to germ-free mouse EVs treated with sitagliptin, which suggests that BEVs likely contain enzymes with non-specific dipeptidase activity. As expected, the PTP deletion mutant was not affected by DPP-4 activity like that seen in the wild-type Bt conventionalised animals (Figure 3a). Sitagliptin inhibited the activity in mouse EVs, decreasing it to the threshold of detection. The results also showed that BEVs produced in vivo carried active PTP (Figure 3b), with the activity significantly decreased in BEVs from mice colonised with the PTP deletion mutant.

It should be noted that EVs produced in non-colonised germ-free mice degraded PTP substrates, with the activity being approximately two-fold lower than that for BEVs from mice conventionalised with wild-type Bt or the Bt DPP-4 deletion mutant. Therefore, the activity measured in the mixture of caecal BEVs and EVs included both microbial and mouse-derived activity, with each accounting for approximately half of the measured values. As expected, the PTP activity measured for BEVs originating from mice conventionalised with the Bt DDP-4 mutant was similar to the activity in BEVs from mice conventionalised with wild-type Bt (Figure 3b). In contrast, the activity in caecal BEVs and EVs from mice conventionalised with the Bt PTP mutant was lower than that in EVs from non-conventionalised germfree mice, which may be explained by the reduced number of EVs produced in the Bt conventionalised animals or lower PTP activity in EVs produced in Bt conventionalised mice.

### 2.5. DPP-4 Activity in Intestinal-Derived BEVs Is Decreased Under Fasting Conditions

Our proteomics studies showed that the abundance of nine out of ten serine proteases was reduced in BEVs produced in the mouse intestine following 16 h of fasting. To validate these results, we measured the DPP-4 activity in EVs collected from the caecal contents of mice mono-conventionalised with wild-type Bt after fasting for 5 h. As a comparator, we measured the activity of asparaginase BT_2757, the abundance of which was increased two-fold in BEVs from fasted mice (Appendix A). The activity of DPP-4 was reduced approximately two-fold to levels comparable with those measured for EVs from non-fasted germ-free mice (Figure 4a). The increased asparaginase activity in the BEVs of fasted mice was also confirmed (Figure 4b).

### 2.6. Proteomes of Host EVs Produced in the GIT

Extracellular vesicles from the caeca of mice mono-conventionalised with Bt consisted of a combination of BEVs and host-derived EVs. The latter were composed primarily of exosomes as a consequence of the isolation procedure involving centrifugation (cell pelleting) and two steps of filter sterilisation (with 0.22-µm-pore-size filters), which reduces the presence of microvesicles and apoptotic bodies. By running a peptide match against the UniProt mouse protein database, 967 proteins from mouse EVs were identified with high confidence (Appendix A). Using PCA, the first two PCs accounted for 28.1% and 20.5%, respectively, of the total variation in the dataset, allowing a distinction between the host EV proteomes of fasted and fed mice (Figure 5a). A volcano plot of the dataset showed that the abundance of 64 proteins was significantly increased (>1.3-fold, *p* ≤ 0.05) in fasted conditions in the proteomes of EVs from intestinal cells, whereas the abundance of 108 proteins was significantly reduced (Figure 5b).

The protein list included 64% of the top 100 protein markers identified in EVs (http://microvesicles.org/extracellular_vesicle_markers, accessed on 14 October 2024) [39] (Appendix A). These comprised tetraspanin proteins belonging to a family of membrane proteins [40,41], including the cell surface glycoproteins CD63, CD82 and CD9 (included in the list despite a 5% FDR value); the endosomal sorting complex required for transport (ESCRT) proteins TSG101 and ALIX; integrins such as Itgb1; actin (Actb); heat shock proteins including HSP90; six members of the 14-3-3 protein family, comprising phospho-binding proteins; and nine annexins (Anxa1-7, -11 and -13), all commonly found in eukaryotic vesicles [42].

An overrepresentation analysis was performed to determine whether known biological functions were enriched in mouse EVs compared to mouse genome predicted proteins (Appendix A). Activities related to cellular metabolism were highly enriched, such as cyanamide hydratase activity (GO:0018820) or alkaline phosphatase activity (GO:0004035) (40- and 30-fold enrichment, respectively). Similarly, aminopeptidase and exopeptidase (7.78- and 6.6-fold enrichment, respectively) and metalloexopeptidase and carboxypeptidase activities (6.6- and 6.51-fold enrichment, respectively) were all significantly enriched (*p* = 1 × 10^−4^–8.7 × 10^−8^). By comparison, activities related to cell signalling and DNA transcription were essentially less enriched in the extracted host EVs.

### 2.7. Increased Abundance of Serine Protease Inhibitors and Antimicrobial Proteins in Host EVs Produced In Vivo

When comparing the abundance ratios for each protein in caecal EV preparations derived from fasted versus fed animals (Table 2), three serine protease inhibitors (serpin A3M, A3K and A1E) were more abundant in fasted mice (5-, 2.5- and 2.1-fold, respectively). Serpins act as serine protease inhibitors, representing up to 2–10% of the proteins in human blood, and are the third most common protein family [43]. We also observed a 3.5-fold increase in the abundance of the murine-specific α-defensin CRISC-2 in EVs produced in fasted mice. α-Defensins are produced by Paneth cells of the small intestinal epithelium, acting as part of the innate immune system, with antimicrobial activity against a wide variety of organisms [44].

### 2.8. Potential Interactions Between Host Serpins and Bacterial Serine Proteases

To address the question of whether the three serpins A3M, A3K and A1e, exhibiting increased abundance in the EVs of fasted mice (Table 2), could potentially inhibit the nine serine proteases showing decreased abundance in the BEVs of the same mice (Table 1), we performed structural predictions. The analysis highlighted that the nine bacterial proteases could potentially establish complexes with the serpins A3M, A3K and A1E. The 3D protein structure prediction tool AlphaFold2 assigned pLDDT (high/low/very low) and pDockQ scores to assess the confidence of the predicted protein–protein interactions, with values above 0.5 considered to indicate high-confidence interactions. The average values for each score were calculated based on the five predicted models (Appendix A).

Members of the S9 peptidase family demonstrated the most robust interactions with host serpins, which were characterised by high pDockQ scores and high pLDDT values. In contrast, the serine proteases belonging to the CLP, S41 and PDZ-containing protease families generally exhibited medium- to very-low-confidence interactions, suggesting a lower likelihood of significant connections.

To further explore the high-confidence interactions, these host–microbe protein complexes were visualised using Mol* RCSB 3D Viewer to identify the bacterial S9 protease domain and the reactive loops and bonds on the serpins (Figure 6), with these interactions typically occurring in protease–serpin complexes [45]. The results confirmed that serpins A3K and A1E may interact with the four serine proteases carrying the S9 protease domain (BT_4193, BT_3254, BT_0587, BT_1838), as their reactive bonds are targeted by this domain.

## 3. Discussion

Our study provides novel insights into microbe–host interactions in the mammalian GIT, describing how nutrient stress and a period of fasting affect the composition of bacterial and intestinal EVs. Using the model human commensal gut bacterium Bt, we have shown that the profile of BEV proteins and that of host cell-derived EVs is influenced by nutrient availability, demonstrating the selective enrichment of proteins and enzymes capable of modulating host metabolism.

Fasting has been shown to increase the relative abundance of Bacteroidetes in the GIT across various animal models [46,47], likely due to their ability to utilise host glycans as alternative energy sources when dietary nutrients are scarce [48,49,50]. Consistent with this observation, our findings revealed that fasting conditions led to selective changes in BEV proteomes. Notably, we did not observe an increase in host glycan-specific and surface-exposed glycohydrolases in BEVs from fasted animals. Indeed, their abundance exhibited a downward trend of up to two-fold compared to their levels in BEVs from fed animals. However, it is interesting to note that, for the PULs identified, the abundance of the integral membrane oligosaccharide importer SusC was increased by about 50% in BEVs produced under fasting conditions, whereas other proteins belonging to the same PULs, including SusD (nutrient binding accessory protein), were equally or less abundant under fasting conditions. The 50% increase in SusC likely reflects an adaptive mechanism employed by Bt to enhance its capacity to utilise host-derived glycans as an alternative energy source during nutrient scarcity.

Our proteomics analysis revealed a two-fold reduction in BT_4193’s abundance in BEVs from fasted mice, alongside decreased levels of nine out of ten serine proteases, including PTP. The reduction in DPP-4 activity observed in fasted mice is primarily attributable to decreased bacterial DPP-4 abundance, as the host DPP-4 levels detected in EVs remained unchanged (Appendix A). This was confirmed by measuring the activity of DPP-4 and PTP in fasted versus fed mice. Proteases released by commensal bacteria play a critical role in maintaining the balance between microbial proteases and host targets, with host protease inhibitors being essential in preserving GIT integrity. The disruption of this equilibrium is linked to diseases such as ulcerative colitis (UC), where elastase-like serine protease activity correlates with *Phocaeicola vulgatus* abundance and increased inflammation following microbiota transfer [51,52,53,54]. The observed reduction in serine proteases and possibly metalloproteases in BEVs from fasted mice may reflect a coordinated mechanism to preserve intestinal homeostasis under nutrient stress. This is further supported by the increased levels of protease inhibitors in host EVs.

The observed reduction in bacterial serine proteases within BEVs during fasting may be explained by two complementary mechanisms. First, this downregulation likely reflects an adaptive response by Bt to nutrient scarcity, prioritising the expression of proteins involved in alternative energy acquisition, such as the increased abundance of the oligosaccharide importer SusC over energetically costly protease production. Second, it is also possible that host-derived factors contribute to this regulation. For example, the increased presence of host serine protease inhibitors (serpins) and antimicrobial peptides like CRISC-2 α-defensin in host EVs during fasting could signal to the microbiota, either directly or indirectly, to suppress protease synthesis or alter the EV cargo composition. Such host–microbe crosstalk may serve to further protect the gut epithelium from excessive proteolytic activity under conditions of nutrient stress, thereby maintaining intestinal homeostasis. Together, these mechanisms highlight the dynamic and coordinated interplay between the host and microbiota in response to environmental changes.

BT_4193 may influence host physiology by affecting protein and glycan (e.g., gluten) digestion, signal transduction and apoptosis [32]. Given the ability of Bt BEVs to access systemic circulation [55,56], it is plausible that BEVs impact host metabolism, immune function and behaviour by cleaving signalling molecules such as incretins, growth factors, cytokines and neuropeptides. For example, bacterial DPP-4 in the GIT can degrade GLP-1, impairing blood glucose homeostasis in mice with a leaky gut [38]. Even in a healthy GIT with an intact epithelial barrier, BEVs carrying DPP-4 may influence glucose regulation by inactivating GLP-1, a hypothesis that warrants further investigation.

We have also shown that BT_1314 shares structural and functional similarities with YkfC of Gram-positive Bacilli, a dipeptidyl peptidase that cleaves the bonds between glutamate and DAP residues in murein peptides derived from PG recycling [35]. Using LC-MSMS, BT_1314 was shown to hydrolyse the pro-inflammatory tripeptide Tri-DAP, an agonist of the NOD1 receptor, and its signalling pathway [37]. The activity of BT_1314 in BEVs may help to counteract inflammatory responses triggered by PG murein peptides released by intestinal bacteria, potentially preventing local inflammation in the intestinal mucosa.

The protein profile of mammalian EVs produced in the caeca of fasted mice contained three serine protease inhibitors, A3M, A3K and A1E serpins (Figure 6), with their abundances increased two- to five-fold compared to fed mice. In humans, the serpin A3 protein is produced primarily by the liver; it is secreted into the plasma and contributes to anti-inflammatory and antiviral responses [57]. As the mouse A3M and A3K serpin protein sequences are only partially similar to that of human serpin A3 (58% and 60% amino acid sequence identities, respectively), and due to the limited functional data in mice, it is difficult to predict their roles [57]. Moreover, seven additional serpins (InterPro family IPR000215) were identified, exhibiting similar abundances in fasted and fed mice (Appendix A). High protease activity measured in the faeces of patients suffering from irritable bowel syndrome has been shown to correlate with a decrease in microbial diversity [58]. It is therefore speculated that the various serine protease inhibitors detected in EVs produced in Bt mono-colonised germ-free mice are a consequence of the lack of microbial diversity and could counteract the detrimental effects of proteases produced in high abundance by a *Bacteroides* species [53] in the GITs of these mice.

The structural analysis of bacterial serine protease–host serpin interactions revealed the mechanisms involved in maintaining gut homeostasis. The peptidase S9 domain exhibits high affinity for serpin-mediated inhibition, suggesting a critical role in regulating proteolytic activity due to its broad substrate specificity. These proteases degrade dietary and host-derived proteins [59], while host serpins (e.g., serpin A3K/A1E) likely limit excessive activity through reactive centre loop (RCL) interactions, forming covalent inhibitory complexes [60] to prevent gut epithelial damage. Computational predictions (e.g., AlphaFold-Multimer) highlight potential RCL-mediated binding; however, experimental validation is required to confirm these interactions. SDS-PAGE can verify covalent serpin–protease complexes via distinct migration shifts due to their SDS-stable nature, while protease inhibition assays can assess the functional suppression of proteolytic activity. Future studies integrating these approaches will clarify the physiological relevance of these interactions in intestinal homeostasis.

The human and mouse small intestines share many similarities in their microbial defence strategies, including the production of α-defensins, which are also found in EVs [61]. Mice, however, produce a unique antimicrobial peptide and member of the CRS (cryptdin-related sequences) peptide family that are not found in humans [62]. We observed a 3.5-fold increase in the abundance of a member of the CRS family, CRISC-2 α-defensin, in EVs produced in fasted mice. Whether a decrease in nutrient availability in the mouse intestine leads to the increased expression of CRISC-2 or to an increased number of the secretory Paneth cells and/or to CRISC-2 preferentially packaged into EVs needs to be determined.

The heterogeneity of mammalian EVs complicates the interpretation of data obtained from complex biological tissue such as the GIT samples used in the present study. Clearly, there is a need to obtain pure preparations of different EV populations, including both mammalian and bacterial EVs. One promising approach to EV separation involves high-resolution density gradient fractionation to separate small EVs (sEVs) from non-vesicular material, followed by the separation of exosomes from other EVs using direct immunoaffinity capture [42]. This approach resulted in the isolation of vesicles carrying the tetraspanin exosomal markers CD9 and CD63, which were identified in the present study, whereas CD81 was not. Some proteins, including GAPDH, ENO1, 14-3-3, HSP90 and PARK7/DJ1, identified in our study, were present in sEVs but absent in classical exosomes [42]. Similarly, cytoskeleton proteins like actins, tubulins and keratins, found in our EV samples, were absent in classical exosomes and present in sEV proteomic datasets [42]. Other discriminatory markers identified in varying abundances in our EV samples included membrane-bound annexins, such as annexin A1, which is characteristic of microvesicles, while others may be associated with exosomes or sEVs [43]. Additionally, the ESCRT proteins TSG101 and ALIX were identified, with ALIX being strongly associated with classical exosomes [42].

In summary, our findings provide evidence for the influence of nutrient stress on the protein composition of BEVs and intestinal EVs in the mouse caecum, highlighting the dynamic interplay and interactions between the host and its microbiome. This manifests at the level of the cross-kingdom regulated production of bacterial proteases, countered by the production of specific inhibitors by host cells in the GIT that can help to maintain intestinal homeostasis and GIT health.

## 4. Materials and Methods

### 4.1. Animal Studies

All mice were maintained under a 12 h light/dark cycle and received autoclaved water and an RM3 (Autoclavable) (GF) diet (Special Diets Services). Animal experiments were conducted in full accordance with the Animal Scientific Procedures Act 1986 under UK Home Office approval and following approval by the local Animal Welfare and Ethical Review Body (AWERB; approval code: P723E9201, approval date: 14 April 2020). No explicit exclusion criteria were set for animals a priori, but any animals showing signs of distress or illness would have been removed based on standard animal welfare protocols. For the proteomics analysis, ten germ-free C57BL/6 (males, 14 weeks old) mice were gavaged with 108 CFU Bt in 100 μL PBS. Mice had unrestricted access to chow and water for 2 days, after which a group of 5 mice was fasted for 16 h. For the validation study, BEV enzymatic activity ex vivo was measured after the fasting of the animals. Germ-free C57BL/6 mice (n = 5 per group), gavaged with Bt as described above, also had unrestricted access to chow and water for 2 days, with groups of mice fasted for 5 h. At post-mortem, the caecal contents were collected, homogenised in PBS (10% *w*/*v*) and centrifuged for 2 min at 100× *g* and the supernatant collected. A 100 μL aliquot was removed to enumerate bacteria on BHI-haemin agar (=12 ± 3 × 1010 CFU/g colon content). The supernatants were then centrifuged at 5500× *g* and 4 °C for 15 min and filtered through polyethersulfone (PES) membranes (0.22 μm pore size) (Sartorius, Göttingen, Germany) to remove debris and remaining cells. Vesicle suspensions were concentrated by crossflow ultrafiltration (100 kDa MWCO, Vivaflow 200, Sartorius) to 5 mL, rinsed via the addition of 500 mL of PBS, pH 7.4, and concentrated again by crossflow filtration to 5 mL, and the retentates were concentrated to 1 mL with a Vivaspin 20 centrifugal concentrator (100 kDa molecular weight cut-off, Sartorius). Further purification of the mixture of BEVs and EVs was performed by size exclusion chromatography (SEC) using either CL2-B Sepharose (Sigma-Aldrich, St. Louis, MO, USA) in PBS buffer for the proteomics analysis, as previously described [21], or using a qEV/35 nm series SEC column according to the manufacturer’s instructions (IZON Science, Lyon, France), with the pooled collected fractions filtered through PES membranes (0.22 μm pore size) and stored at 4 °C. For the proteomics analysis, the pooled fractions were adjusted to 8.9 mL and the BEV suspension ultracentrifuged (150,000× *g* at 4 °C or 2 h in a Ti70 rotor (Beckman Instruments, Brea, CA, USA)). After ultracentrifugation, the supernatant was removed using a vacuum pump and the vesicle pellets snap-frozen in liquid nitrogen and stored at −80 °C prior to extraction.

### 4.2. Nanoparticle Analysis

For the BEV preparations used in the proteomics studies, the hydrodynamic size distribution of the vesicles was performed on aliquots of BEV suspensions diluted 100-fold with PBS. Videos were generated using a Nanosight nanoparticle instrument (NanoSight Ltd., Malvern Panalytical, Malvern, UK) to count BEV numbers. A 1 min AVI file was recorded and analysed using the NTA (Version 2.3 Build 0011 RC, Nanosight) software to calculate the size distributions and vesicle concentrations using the following settings: calibration—166 nm/pixel; blur—auto; detection threshold—10, minimum track length—auto, temperature—21.9 °C, viscosity—0.96 cP. The accuracy of the measurement was confirmed using 100 nm silver nanoparticles (Sigma-Aldrich).

For the BEV preparations used in enzyme assays, the size and concentration of the isolated Bt BEVs were determined using a ZetaView PMX-220 TWIN instrument according to the manufacturer’s instructions (Particle Metrix GmbH, Inning am Ammersee, Germany). Aliquots of BEV suspensions were diluted 1000- to 20,000-fold in particle-free PBS or water for analysis. Size distribution video data were acquired using the following settings: temperature—25 °C; frames—60; duration—2 s; cycles—2; positions—11; camera sensitivity—80; and shutter value—100. Data were analysed using the ZetaView NTA software (version 8.05.12) with the following post-acquisition settings: minimum brightness—20; max area—2000; min area—5; and trace length—30.

### 4.3. Proteomics

Comparative proteomics was carried out on samples of BEVs and EVs isolated from the caeca of fed or fasted mice. Five mice were used for each condition, providing 10 datasets, including ratios (fasted versus fed) for each protein identified, with the level of confidence determined by the false discovery rate (FDR); these were then further analysed. Proteomic samples were prepared and analysed by the Bristol University Proteomics Facility.

#### 4.3.1. TMT Labelling and High-pH Reversed-Phase Chromatography

First, 50 ug of each sample was digested with trypsin (1.25 µg trypsin; 37 °C, overnight) and labelled with Tandem Mass Tag (TMT) ten plex reagents according to the manufacturer’s protocol (Thermo Fisher Scientific, Loughborough, LE11 5RG, UK), and the labelled samples were pooled.

A 100 ug aliquot of the pooled sample was evaporated to dryness, resuspended in 5% formic acid and then desalted using a SepPak cartridge according to the manufacturer’s instructions (Waters, Milford, MA, USA). The eluate from the SepPak cartridge was again evaporated to dryness and resuspended in buffer A (20 mM ammonium hydroxide, pH 10) prior to fractionation by high-pH reversed-phase chromatography using an Ultimate 3000 liquid chromatography system (Thermo Scientific). In brief, the sample was loaded onto an XBridge BEH C18 column (130Å, 3.5 µm, 2.1 mm × 150 mm, Waters, UK) in buffer A and peptides eluted with an increasing gradient of buffer B (20 mM ammonium hydroxide in acetonitrile, pH 10) from 0 to 95% over 60 min. The resulting fractions (concatenated to 15 in total) were evaporated to dryness and resuspended in 1% formic acid prior to analysis by nano-LC MSMS using an Orbitrap Fusion Tribrid mass spectrometer (Thermo Scientific).

#### 4.3.2. Nano-LC Mass Spectrometry

The high-pH RP fractions were further fractionated using an Ultimate 3000 nano-LC system in line with an Orbitrap Fusion Tribrid mass spectrometer (Thermo Scientific). In brief, peptides in 1% (vol/vol) formic acid were injected onto an Acclaim PepMap C18 nano-trap column (Thermo Scientific). After washing with 0.5% (vol/vol) acetonitrile and 0.1% (vol/vol) formic acid, peptides were resolved on a 250 mm × 75 μm Acclaim PepMap C18 reverse-phase analytical column (Thermo Scientific) over a 150 min organic gradient, using 7 gradient segments (1–6% solvent B over 1 min, 6–15% B over 58 min, 15–32% B over 58 min, 32–40% B over 5 min, 40–90% B over 1 min, held at 90% B for 6 min and then reduced to 1% B over 1 min) with a flow rate of 300 nL min^−1^. Solvent A was 0.1% formic acid and solvent B was aqueous 80% acetonitrile in 0.1% formic acid. Peptides were ionised by nano-electrospray ionisation at 2.0 kV using a stainless-steel emitter with an internal diameter of 30 μm (Thermo Scientific) and a capillary temperature of 275 °C.

All spectra were acquired using an Orbitrap Fusion Tribrid mass spectrometer controlled by the Xcalibur 2.1 software (Thermo Scientific) and operated in data-dependent acquisition mode using an SPS-MS3 workflow. FTMS1 spectra were collected at a resolution of 120,000, with an automatic gain control (AGC) target of 200,000 and a maximum injection time of 50 ms. Precursors were filtered with an intensity threshold of 5000, according to the charge state (to include charge states 2–7), and with monoisotopic peak determination set to peptide. Previously interrogated precursors were excluded using a dynamic window (60 s ± 10 ppm). The MS2 precursors were isolated with a quadrupole isolation window of 1.2 *m*/*z*. ITMS2 spectra were collected with an AGC target of 10,000, maximum injection time of 70 ms and CID collision energy of 35%.

For the FTMS3 analysis, the Orbitrap mass spectrometer (Thermo Fisher Scientific, Bremen, Germany) was operated at a 50,000 resolution with an AGC target of 50,000 and a maximum injection time of 105 ms. Precursors were fragmented by high-energy collision dissociation (HCD) at a normalised collision energy of 60% to ensure the maximal TMT reporter ion yield. Synchronous precursor selection (SPS) was enabled to include up to 10 MS2 fragment ions in the FTMS3 scan.

#### 4.3.3. Data Analysis

The raw data files were processed and quantified using the Proteome Discoverer software v2.1 (Thermo Scientific) and searched against the UniProt Mus musculus database or the Bt database using the SEQUEST HT algorithm. Peptide precursor mass tolerance was set at 10 ppm, and MS/MS tolerance was set at 0.6 Da. The search criteria included the oxidation of methionine (+15.995 Da), acetylation of the protein N-terminus (+42.011 Da) and methionine loss plus acetylation of the protein N-terminus (–89.03 Da) as variable modifications and the carbamidomethylation of cysteine (+57.021 Da) and the addition of the TMT mass tag (+229.163 Da) to peptide N-termini and lysine as fixed modifications. Searches were performed with full tryptic digestion, and a maximum of 2 missed cleavages were allowed. The reverse database search option was enabled, and all data were filtered to satisfy a false discovery rate (FDR) of 1%. The abundance values for each TMT channel were normalised so that all channels had the same total abundance.

### 4.4. Statistical Analysis

PCA was performed using the R-based statistical analysis tool provided by the MetaboAnalyst 6.0 platform (https://www.metaboanalyst.ca/, accessed on 15 February 2025). The normalised dataset of relative abundances obtained from Proteome Discoverer v2.1 for each protein in each condition (5 replicates) was analysed. The data were log-transformed, and 2D score plots with PC1 on the X axis and PC2 on the Y axis, including the 95% confidence ellipses, were generated.

A volcano plot combining a fold change analysis (1.3 threshold) and an unpaired *t*-test (*p*-value threshold 0.05) was also generated.

For the enzyme assays, data were subjected to one-way ANOVA followed by Dunnett’s multiple comparison post hoc test using the GraphPad Prism 5 software. Statistically significant differences between two mean values were established by adjusted *p*-values. Data are presented as the mean ± standard deviation.

### 4.5. Proteomics Data Curation

For the Bt BEV protein profile, the raw results displayed a list of 572 proteins identified in BEVs produced in fasted and fed mice. Using the 99% confidence level (1% FDR), 34 proteins were removed. Proteins annotated as “not found in sample” in more than 50% of the total samples for each group (fasted or fed) were removed, resulting in a total of 528 identified proteins.

For the EV protein profile, the raw results displayed a list of 1189 proteins identified in BEVs produced in fasted and fed mice. Using the 99% confidence level (1% FDR), 159 proteins were removed. Proteins annotated as “not found in sample” in more than 50% of the total samples for each group (fasted or fed) were removed, resulting in a total of 967 identified proteins.

### 4.6. Gene Ontology Analysis

Protein classification according to species-specific gene ontology (GO) annotations and overrepresentation analysis were performed using PANTHER version 14.0 (http://www.pantherdb.org/, accessed on 18 July 2024) [63]. For the functional analysis in Bt, the source data included Bt BEV-derived proteins, which were compared to the entire Bt genome. For the mouse data, the proteins identified in EVs were compared to the whole mouse genome. The overrepresentation analysis was conducted using Fisher’s exact test to determine the significance of annotation overrepresentation.

### 4.7. Electron Microscopy

For negatively stained TEM images, BEV and EV pellets were resuspended in 100 µL of d.H2O by vortex. Then, 10 µL of each suspension was transferred to Cu 200 Formvar/carbon grids (Agar Scientific, Stansted, UK), avoiding large aggregates of material. The suspension was left on the grid for 1 min before wicking off excess liquid with filter paper. The grids were stained with 2% aqueous uranyl acetate solution for 1 min, followed by wicking off excess stain with filter paper and being left to dry thoroughly. The grids were examined and imaged using a Tecnai G2 20 twin-transmission electron microscope (Thermo Fisher Scientific, Eindhoven, The Netherlands) at 200 kV.

### 4.8. BT_3254 (PTP) and BT_4193 (DPP-4) Deletion Mutants

BT_3254 and BT_4193 deletion mutant strains were generated using the method developed by Garcia-Bayona and Comstock [64]. Briefly, BT_3254 and BT_4193 flanking DNA fragments containing an overlap sequence were generated (upstream primers: BT3254-1 and -2 and downstream primers: BT3254-3 and -4; upstream primers: BT4193-1 and -2 and downstream primers: BT4193-3 and -4; Appendix A). Recombinant PCR was used to combine the fragments. BamHI and EcoRV sites were introduced during amplification, flanking the fragment to allow for ligation into pLGB13 [64]. E. coli PIR1+ (Thermo Fisher Scientific) was used to carry out the cloning and as the donor strain. Bt VPI-5482 was conjugated using triple mating, with *E. coli* HB101 (pRK2013) used as the helper strain [65]. Transconjugants were selected on BHIS agar (gentamicin (200 μg/mL); erythromycin (25 μg/mL)) after 48 h. Two colonies were grown overnight in BHIH (gentamicin (200 μg/mL); erythromycin (25 μg/mL)) and streaked out on the counter selection plate of BHIH agar (gentamicin 200 μg/mL, anhydrotetracycline 100 ng/mL). Colonies lacking the gene were selected by colony PCR. Positive colonies were grown in BHIH (gentamicin (200 μg/mL); anhydrotetracycline (100 ng/mL)) and stored at 80 °C in 10% glycerol in BHIH prior to use.

### 4.9. DPP-4, PTP and Asparaginase Assays

BEV and EV suspensions were prepared as described above. DPP-4 assays were performed as previously described [21]. Briefly, 75 µL of 50 mM Tris–HCl buffer (pH 7.5) and 5 µL of Ala-Pro-pNA substrate (Bachem, 5 mg/mL in methanol) were added to 20 µL BEV/EV suspensions in a 96-well microtiter plate. For the PTP assays, the same procedure as above was carried out using Ala-Phe-Pro-pNA (Bachem, 5 mg/mL in methanol) as the substrate. Increasing concentrations of p-nitroaniline (Abcam) were used to establish a standard curve. The reaction mixture was incubated at 37 °C, and the absorbance at 405 nm was measured at 1 min intervals for 100 min using a FLUOstar Omega (BMG Labtech, Ortenberg, Germany) plate reader.

Asparaginase activity was measured using the Nessler reagent method. Briefly, 10 μL of asparagine 189 mM was added to 100 μL of buffer (50 mM Tris buffer, pH 8.6) and 50 μL of water, to which 40 μL of the BEV/EV suspension was added. The reaction mix was incubated at 37 °C, and 20 μL of the reaction mix was collected every 30 min, to which 1 μL of 1.5 M trichloroacetic acid was added. The mix was centrifuged for 2 min at 20,000× *g* and the supernatant added to 430 μL of water. Then, 50 μL of Nessler reagent (Camlab, Cambridge, UK) was added to the mix, and the absorbance at 436 nm was recorded for standards, tests and blanks using a FLUOstar Omega (BMG Labtech) plate reader. To establish a standard curve, dilutions of (NH4)_2_SO_4_ (25–100 mM) were included. The concentration of protein in BEVs/EVs was determined using the Qubit protein assay (Thermo Fisher Scientific, Waltham, MA, USA) and was consistent with the concentrations of particles determined by ZetaView.

### 4.10. Recombinant BT_1314 and Enzymatic Activity

The 405-amino-acid product of the BT_1314 C-terminal region, excluding the 21-amino-acid N-terminal predicted signal peptide, was purified using the His-Tag technique. A PCR fragment was generated using the primer pair BT1314_up and BT1314_down (Appendix A), and the resulting fragment was cloned into the NdeI/BamHI restriction sites of the pET-15b expression vector (Novagen, Madison, WI, USA), which carries an N-terminal His-Tag sequence. The resulting plasmid was used to transform BL21CodonPlus(DE3)-RIL *E. coli* cells (Stratagene, La Jolla, CA, USA). The Ni-NTA Fast Start Kit (Qiagen, Venlo, The Netherlands) was used to purify the protein, according to the manufacturer’s instructions. Cells were grown at 30 °C for 16 h with a concentration of 0.5 mM IPTG. The imidazole used to elute the protein was removed using a PD-10 desalting column (Amersham Biosciences, Little Chalfont, UK) and the buffer exchanged with 50 mM Tris–HCl, 300 mM NaCl, pH 7.2. The protein concentration was determined by direct UV measurement at 280 nm and by using the Bradford method.

To assess the activity of the recombinant BT_1314, 50 ng of the purified enzyme was added to 50 μM of Ala-γ-D-Glu-DAP (Tri-DAP, Eurogentec, Seraing, Belgium) in a reaction volume of 50 μL buffered with 50 mM Tris–HCl (pH 7.5) and incubated for 16 h at 37 °C. After the addition of 50 μL acetonitrile, the samples were centrifuged at 17,000× *g* for 5 min. The supernatant was analysed by LC-MS on a Raptor Polar x (2.7 µm, 100 × 2.1 mm (Restek), using 0.5% formic acid in 20 mM ammonium formate (A) and 0.5% formic acid and 20 mM ammonium formate in 90% acetonitrile as the mobile phase, using the following gradient: isocratic elution with 5% A for 5 min, followed by a linear increase in the concentration of A to 50% over three minutes. The concentration was retained at 50% for 5 min before the column was reconditioned with 5% A for 4 min. The analytes were detected by a Xevo TQ Absolute triple-quadrupole mass spectrometer (Waters, Milford, MA, USA) using selected ion monitoring. Mass values of 189.1, 217.2 and 389.1, corresponding to DAP-1, L-Ala-γ-D-Glu and Tri-DAP, respectively, were monitored in negative ion mode, with a cone voltage of 25 V and a dwell time of 0.003 s.

### 4.11. Structural Analysis of Potential Bacterial Protease—Host–Serpin Interactions

AlphaFold2 [66], a deep learning-based software program that predicts 3D protein structures, was used to study potential host–microbe interactions between the nine BEV-derived serine proteases and the three serpin proteins (serpin A3M, A3K and A1E) from mouse vesicles. The sequences of the bacterial protease and host serpin were obtained from the UniProt database and served as the input for the tool. Multiple sequence alignments (MSA) were generated using the MMseqs2 algorithm within the neurosnap platform [67], identifying homologous sequences from the databases. This approach enhances the quality of sequence alignments, providing a more diverse input for structural predictions. AlphaFold2-Multimer (version 3) [68] was employed to predict the three-dimensional structures of the protease–serpin complexes. The model was configured to run with the following parameters.

MSA mode: mmseqs2_uniref_env–AlphaFold 2 suggests this mode, as, in most cases, mmseqs2_uniref_env tends to produce the best results.Paired mode: unpaired—as the bacterial and host proteins are from different organisms, the unpaired setup separates the MSA for each chain.Number of ensemble recycles: default settings (five recycles) were used to balance the computational time with model accuracy.Number of ensembles: default settings—1 ensemble. The trunk of the network is run multiple times with different random choices for the MSA cluster centres.

The analysis resulted in five independent multimer models, and their accuracy was evaluated based on the pLDDT (evaluates the local confidence) and PAE (positional error between residue pairs within and between chains) confidence metrics provided by AlphaFold2. The generated models were ranked based on the pLDDT and overall structural stability described by the pDockQ score, which measures the probability of protein–protein interactions. Molecular interactions were visualised in RBCS 3D Viewer (https://www.rcsb.org/3d-view/, accessed on 18 September 2024) [69] using the PDB output from the analysis.

## Figures and Tables

**Figure 1 ijms-26-04080-f001:**
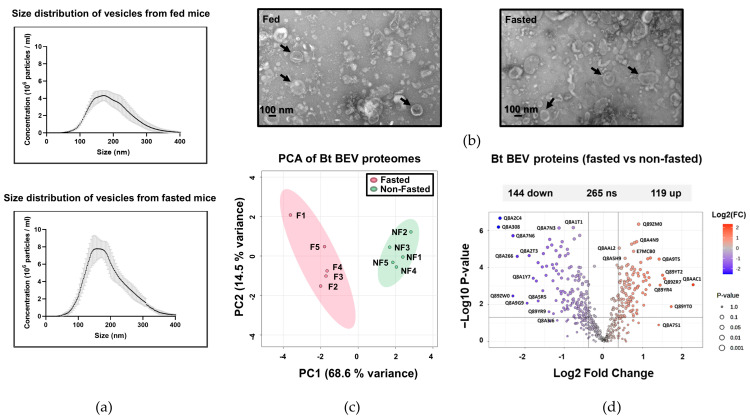
Structure, size, concentration and protein content of BEVs produced in vivo. (**a**) Nanoparticle tracking analysis of BEV suspensions. Points (black) represent the mean and the error bars (grey) represent the standard deviation (SD), n = 3. (**b**) TEM images of vesicles extracted from caecal content of fed and fasted germ-free mice mono-conventionalised with Bt. Scale bar = 100 nm. NF: non-fasted; F: fasted. (**c**) Principal component analysis performed on normalised abundances of each protein under each condition. X and Y axes show principal component 1 and principal component 2, explaining 68.6% and 14.5% of the total variance. Prediction ellipses are such that, with probability 0.95, a new observation from the same group will fall inside the ellipse. NF: non-fasted; F: fasted. (**d**) Volcano plots displaying ratios of protein abundance in fasted versus fed conditions. The set thresholds were 0.05 for the *p*-value (n = 5) and 1.3 for the fold change (FC). Features with >50% missing values were removed. Blue dots indicate proteins that are significantly less abundant when obtained from fasted conditions, and red dots indicate proteins that are significantly more abundant.

**Figure 2 ijms-26-04080-f002:**
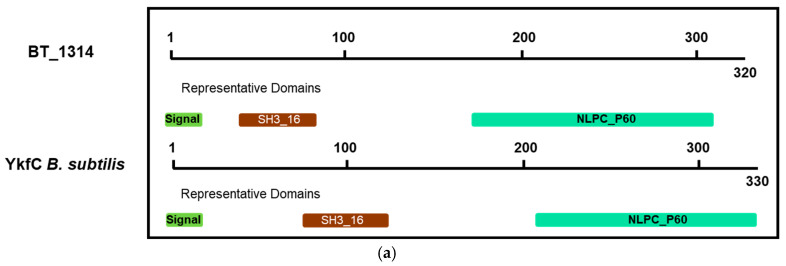
BT_1314 is a murein tripeptide dipeptidase. (**a**) Schematic alignment of BT_1314 and YkfC of *B. subtilis*, including representative conserved domains (Pfam domains). (**b**) LC-MS chromatogram obtained after digestion of Tri-DAP by a recombinant BT_1314 enzyme (rBT_1314). Upper panel: Tri-DAP control; middle panel: DAP standard; lower panel: reaction products obtained after incubation with BT_1314.

**Figure 3 ijms-26-04080-f003:**
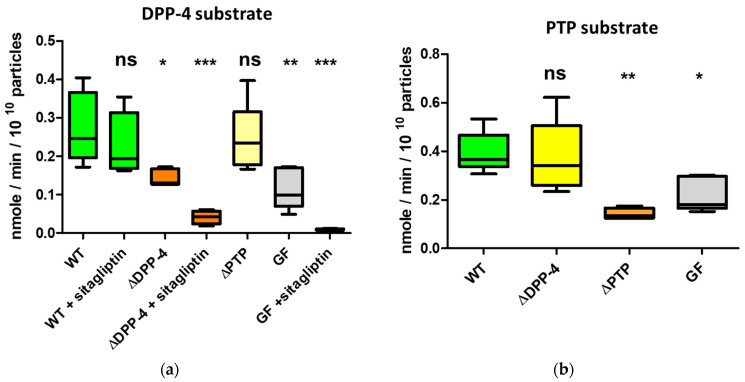
DPP-4 and PTP activity measured in vesicles extracted from caecal contents of mice. (**a**) DPP-4 activity of vesicles extracted from the caeca of germ-free mono-conventionalised mice with wild-type Bt (WT), Bt DPP-4 deletion mutant (ΔDPP-4) and Bt PTP deletion mutant (ΔPTP) (n = 5 each). (**b**) PTP activity of vesicles extracted from the caeca of germ-free mono-conventionalised mice with wild-type Bt (WT), Bt DPP-4 deletion mutant (ΔDPP-4) and Bt PTP deletion mutant (ΔPTP) (n = 5 each). GF = EVs from nonmanipulated germ-free mice. Statistical symbols refer to comparisons of each group to the WT group. ns = not significant; * = *p* ≤ 0.05; ** = *p* ≤ 0.01; *** = *p* ≤ 0.001. Sitagliptin: particles for which 0.33 mM sitagliptin was added to the reaction mixture. Ala-Pro-pNA = DPP-4 chromogenic substrate; Ala-Phe-Pro-pNA = PTP chromogenic substrate.

**Figure 4 ijms-26-04080-f004:**
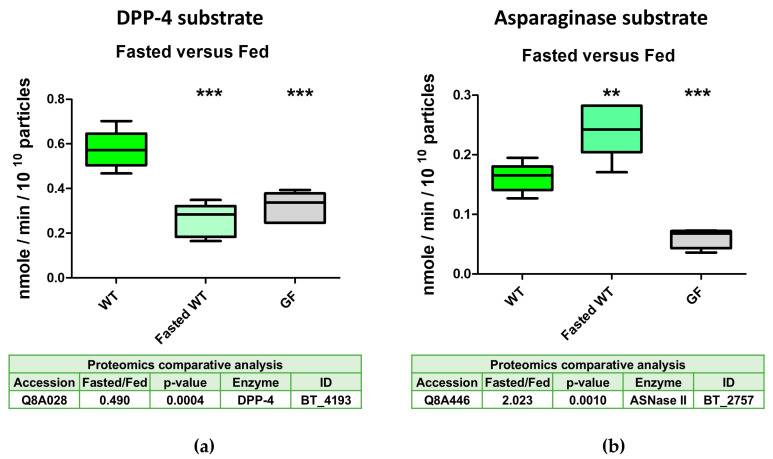
DPP-4 activity is decreased in vesicles extracted from caecal contents of fasted mice mono-conventionalised with Bt WT. (**a**) DPP-4 activity (n = 5); (**b**) ASNase activity (n = 5); GF = EVs from nonmanipulated germ-free mice. Statistical symbols refer to comparisons of each group to the WT group. ** = *p* ≤ 0.01; *** = *p* ≤ 0.001. The results in the two tables are derived from data presented in Appendix A.

**Figure 5 ijms-26-04080-f005:**
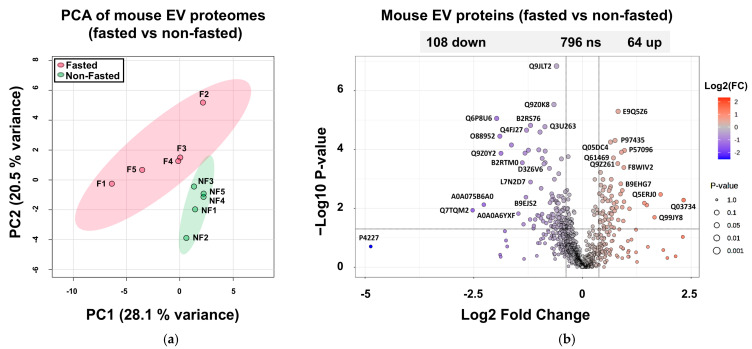
Protein composition of intestinal EVs extracted and purified from the mouse caecum. (**a**) PCA of normalised protein abundances in intestinal EVs under fasted vs. non-fasted conditions. X and Y axes show principal component 1 and principal component 2, explaining 28.1% and 20.5% of the total variance. Prediction ellipses are such that, with probability 0.95, a new observation from the same group will fall inside the ellipse. NF: non-fasted; F: fasted. (**b**) Volcano plots displaying ratios of protein abundance in fasted vs. non-fasted conditions. The set thresholds were 0.05 for the *p*-value (n = 5) and 1.3 for the fold change (FC). Features with >50% missing values were removed. Blue dots indicate proteins that are significantly less abundant when obtained from in vivo conditions, and red dots indicate proteins that are significantly more abundant.

**Figure 6 ijms-26-04080-f006:**
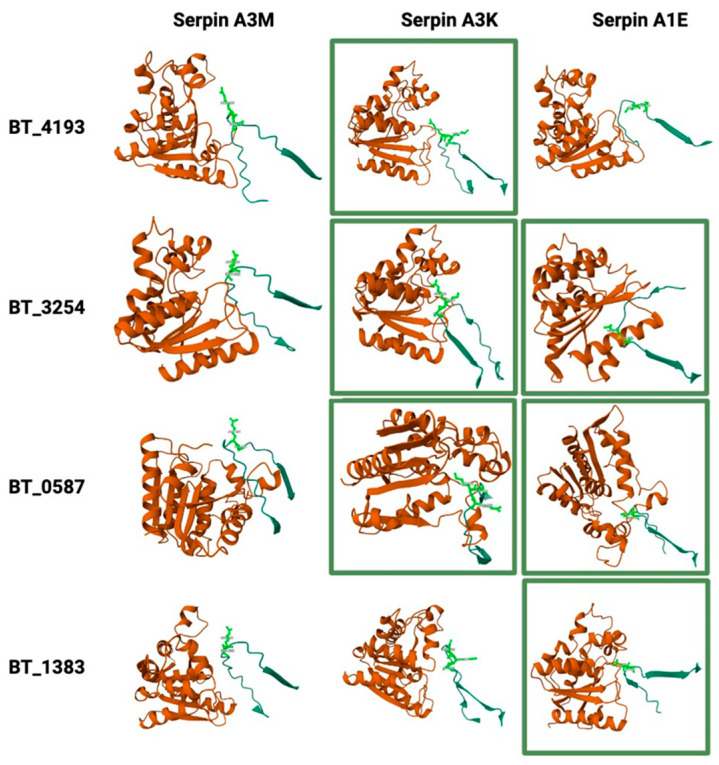
Visualisation of predicted protease–serpin complexes: host–microbe protein complexes were visualised using Mol* RCSB 3D Viewer to identify the bacterial S9 protease domain and the reactive loops and bonds on the serpins. Serpins A3K and A1E are predicted to interact with high confidence (highlighted in green rectangles) with the serine proteases carrying the S9 protease domain (BT_4193, BT_3254, BT_0587, BT_1838), as their reactive sites are targeted by this domain.

**Table 1 ijms-26-04080-t001:** Classes of proteins with decreased abundance in BEVs produced in fasted mice.

Accession	Fasted/Fed ^a^	*t*-Test ^b^	Protein Class ^c^	Accession	Fasted/Fed ^a^	*t*-Test ^b^	Protein Class ^c^
Q8ABE4	0.31	0.0020	Oxidoreductase (PC00176)	Q8A739	0.65	0.0088	Ligase (PC00142)
Q8ABF6	0.34	3.20 × 10^−5^	Serine protease (PC00203) **BT_0154**	Q8A1G3	0.65	5.73 × 10^−5^	Amylase (PC00048)
Q8AAH7	0.35	7.17 × 10^−7^	Carbohydrate kinase (PC00065)	Q8A1H6	0.66	0.0313	Glycosidase (PC00110)
Q89YR9	0.38	0.0077	Lyase (PC00144)	Q8ABF8	0.66	0.0085	Esterase (PC00097)
Q8A3Q9	0.41	4.50 × 10^−5^	Glycosidase (PC00110)	Q8A1Z2	0.67	0.0450	Glycosidase (PC00110)
Q8A2Q1	0.47	0.0005	Serine protease (PC00203) **BT_3254**	Q8AA78	0.67	0.0002	Serine protease (PC00203) **BT_0587**
Q8A3W3	0.48	0.0016	Kinase (PC00137)	Q8A9Q8	0.67	0.2896	Galactosidase (PC00104)
Q8AAV4	0.48	2.59 × 10^−5^	Glycosidase (PC00110)	Q8A7W6	0.68	0.0029	Transporter (PC00258)
Q8A406	0.48	0.0012	Serine protease (PC00203) **BT_2798**	Q8A1F8	0.69	0.0565	Phosphatase (PC00181)
Q8A028	0.49	0.0004	Serine protease (PC00203) **BT_4193**	Q8A1K7	0.70	0.0112	Galactosidase (PC00104)
Q8A8R1	0.50	0.0095	Aldolase (PC00044)	Q89ZQ6	0.72	0.0140	Metalloprotease (PC00153)
Q8A7I6	0.50	0.0042	Hydrolase (PC00121)	Q8A862	0.73	0.0081	Serine protease (PC00203) **BT_1312**
Q8A156	0.50	0.0007	Metalloprotease (PC00153)	Q8A4H0	0.74	0.1614	Hydrolase (PC00121)
Q8A7J8	0.51	0.0016	Isomerase (PC00135)	Q8A123	0.74	0.0165	Chaperone (PC00072)
Q8A4T7	0.51	7.46 × 10^−6^	Dehydrogenase (PC00092)	Q8A4P9	0.76	0.0529	Transferase (PC00220)
Q8A1P7	0.54	2.22 × 10^−5^	Chaperone (PC00072)	Q8A3A0	0.77	0.0715	Dehydrogenase (PC00092)
Q8A0A2	0.56	0.0017	Lyase (PC00144)	Q89ZX8	0.77	0.0165	Dehydrogenase (PC00092)
Q8A5T2	0.57	0.0003	Isomerase (PC00135)	P94598	0.77	0.0438	Dehydrogenase (PC00092)
Q8AAB5	0.58	0.0052	Phosphodiesterase (PC00185)	Q8A0Q9	0.77	0.2888	Hydrolase (PC00121)
Q8A326	0.58	0.0058	Structural protein (PC00211)	Q8A470	0.79	0.0589	DNA polymerase (PC00019)
Q8A414	0.58	0.0006	Kinase (PC00137)	Q8A8Z9	0.79	0.2252	Lyase (PC00144)
Q8A8K6	0.59	0.0007	Cysteine protease (PC00081)	Q8A1Y1	0.80	0.0455	Hydrolase (PC00121)
Q8A5K0	0.59	0.0068	Serine protease (PC00203) **BT_2239**	Q8A0N1	0.80	0.0442	Hydrolase (PC00121)
Q8A1S8	0.61	0.0824	Dehydrogenase (PC00092)	Q8A6B2	0.83	0.0783	Dehydrogenase (PC00092)
Q8A1L2	0.63	0.0080	Protease (PC00190)	Q8A3H7	0.83	0.1219	Chaperone (PC00072)
Q8A3Q6	0.64	0.0103	Glycosidase (PC00110)	Q8A6N9	0.84	0.0733	Serine protease (PC00203) **BT_1838**
Q89ZB2	0.64	0.0056	Voltage-gated ion channel (PC00241)	Q8A129	0.85	0.0885	Serine protease (PC00203) **BT_3842**
Q8AAW1	0.64	0.0002	Isomerase (PC00135)				

^a^ Protein abundance fold change in BEVs from fasted versus fed mice; ^b^
*t*-test column lists *p*-values from statistical comparisons; ^c^ Panther classification (Panther Class ID).

**Table 2 ijms-26-04080-t002:** Differentially abundant proteins in EV preparations from the caeca of fasted versus fed mice.

Accession	Description	(S)/(NS) ^a^	*t*-Test ^b^	Protein Class
** Increased Abundance in Fasted Mice **				
Q03734	Serine protease inhibitor A3M	5.01	0.0136	Protease inhibitor
Q5ERJ0	CRS1C-2 alpha-defensin	3.47	0.0074	Defensin
Q9CPY7	Cytosol aminopeptidase Lap3	2.79	0.0470	Aminopeptidase
A0A0R4J0I1	Serine protease inhibitor A3K	2.68	0.0313	Protease inhibitor
Q00898	Alpha-1-antitrypsin 1–5 (Serpina1e)	2.12	0.0102	Protease inhibitor
Q9CYL5	Golgi-associated plant pathogenesis-related protein 1	2.01	0.0220	-
Q8R000	Organic solute transporter subunit alpha	2.01	0.0177	Transport
** Decreased abundance in fasted mice **				
E9Q7Q0	Mucin-4	0.49	0.0012	Cell–matrix adhesion
I6L958	Igk protein	0.49	0.0235	Immunoglobulin
P02816	Prolactin-inducible protein homologue	0.49	0.0144	-
E9Q035	Uncharacterised protein	0.47	0.0125	Transport/Carrier
B1AWC9	Phosphodiesterase	0.46	0.0339	Phosphodiesterase
Q7TQD7	Myo1b protein	0.46	0.0304	Actin-binding
B2RS76	Carboxypeptidase B1 (tissue)	0.44	2.29 × 10^−5^	Peptidase
Q9CQC2	Colipase	0.44	0.0012	Protein-binding activity modulator
Q9D2R0	Acetoacetyl-CoA synthetase	0.43	0.0077	Ligase
Q64444	Carbonic anhydrase 4	0.42	0.0003	Lyase
Q4FJZ7	Ada protein	0.41	0.0002	Deaminase
L7N2D7	Uncharacterised protein	0.41	0.0205	-
P00688	Pancreatic alpha-amylase	0.40	0.0003	Amylase
B2RTM0	Histone H4	0.38	0.0101	Metalloprotease
Q683Y7	Immunoglobulin heavy chain variable region	0.32	0.0171	Immunoglobulin
A0A075B677	Immunoglobulin kappa variable 4-53	0.32	2.75 × 10^−5^	Immunoglobulin
Q9Z0Y2	Phospholipase A2	0.27	0.0016	Phospholipase
O88952	Protein lin-7 homologue C	0.27	0.0043	Cell junction
Q6P8U6	Pancreatic triacylglycerol lipase	0.26	0.0002	Lipase

^a^ Ratios > 2 or <0.5; ^b^
*t*-test column lists *p*-values from statistical comparisons.

## Data Availability

The mass spectrometry proteomics data have been deposited to the ProteomeXchange Consortium via the PRIDE [75] partner repository with the dataset identifier PXD062737. All other data used to support the findings of this study are included within the article.

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
