# Peer review of "Proteomics of Bacterial and Mouse Extracellular Vesicles Released in the Gastrointestinal Tracts of Nutrient-Stressed Animals Reveals an Interplay Between Microbial Serine Proteases and Mammalian Serine Protease Inhibitors"

_ijms, 2025, doi:10.3390/ijms26094080_

Round 1

Reviewer 1 Report

Comments and Suggestions for Authors

In this manuscript, the authors investigated the effect of host dietary conditions on the protein composition of bacterial extracellular vesicles (BEVs) and host extracellular vesicles (EVs) in the gastrointestinal tract of mouse gavaged with Bacteroides thetaiotaomicron (Bt). Using mass spectrometry, the authors identified multiple serine poteases that are down-regulated in BEVs in fasted animals, along with increased levels of several serine protease inhibitors (sermons) in the host EVs. Overall, this is an interesting topic to study with potential clinical relevance. However, most of the results are correlative, and may or may not have functional significance. Also, the study was done with a single type of bacterium, which hardly reflects the dynamic interactions between host and microbiota.

  1. Just to double check - the purification method used in the manuscript does not separate BEVs from host EVs, correct?
  2. Figure 3a: Why does sitagliptin barely affect the DPP-4 activity in BEVs + EVs from mice with wild-type bt,  while almost completely inhibit it in those from mice with DPP-4 depleted bt? Does this suggest that the host DPP-4 is suppressed by baterial DPP-4? If so, what is the mechanism?
  3. Figure 4a: It is a little misleading to attibute the lower DPP-4 activity upon fasting entirely to the decreased DPP-4 levels in BEVs, since host DPP-4 could also affect the readout.
  4. Figure 6: The modeling is intersting, but as mentioned in discussion, this needs to be verifed by interaction and protease activity studies. Also, is the interactions functionally relevant if the bacterial serine proteases are significantly down-regulated already?
  5. Line 386: ...BT_4393... should be BT_4193..?

Author Response

Reviewer 1

We thank the reviewer for their positive comments regarding the clinical relevance of our study. We acknowledge that many of our results are correlative, which is a limitation of our study design. As suggested, future studies will focus on functional experiments to elucidate the mechanistic significance of these observations. Regarding the use of a single bacterial species, we chose Bacteroides thetaiotaomicron due to its well-established role as a major gut symbiont and its extensively characterised extracellular vesicle production. We agree that studying additional bacterial species would provide a broader understanding of host-microbiota interactions and propose this as an avenue for future research. Despite these limitations, we believe our findings offer valuable insights into nutrient stress effects on BEVs and host EVs, contributing to our understanding of microbe-host communication under fasting conditions.

Comment 1: Just to double check - the purification method used in the manuscript does not separate BEVs from host EVs, correct?

Correct, the purification method used in the study does not fully separate BEVs from host-derived EVs. As described in the manuscript, the isolation protocol (centrifugation, filtration, SEC) results in a mixture of BEVs and host EVs. This has been clarified in the results section (lines 114–116). In an attempt to isolate host EVs from the mixture, we used the EV Isolation Kit Pan, mouse (Miltenyi Biotec), which targets tetraspanin markers (CD9, CD63, CD81) on host EVs. However, this approach cannot guarantee complete separation, as some host EVs lacking these markers may be missed, and some BEVs may persist due to non-specific interactions. This highlights the current technical challenges in achieving complete separation of BEVs from host EVs in complex biological samples.

Comment 2: Figure 3a: Why does sitagliptin barely affect the DPP-4 activity in BEVs + EVs from mice with wild-type bt, while almost completely inhibit it in those from mice with DPP-4 depleted bt? Does this suggest that the host DPP-4 is suppressed by bacterial DPP-4? If so, what is the mechanism?

The observed difference in sitagliptin’s effect on DPP-4 activity between wild-type and DPP-4-depleted Bt mice arises from differential sensitivity of bacterial versus mammalian DPP-4 to the inhibitor, rather than suppression of host DPP-4 by bacterial enzymes. In wild-type Bt mice, BEVs contain bacterial DPP-4 (BT_4193), which has structural variations in its active site that reduce sitagliptin’s binding affinity compared to mammalian DPP-4 (https://doi.org/10.1126/science.add5787 and lines 222-224 of this manuscript). This allows residual bacterial DPP-4 activity to persist even in the presence of sitagliptin, while host-derived DPP-4 in EVs is fully inhibited. In contrast, DPP-4-depleted Bt mice lack bacterial DPP-4, leaving only host-derived DPP-4 activity, which is highly sensitive to sitagliptin. The mechanism hinges on structural divergence between bacterial and mammalian DPP-4 active sites: mammalian DPP-4 has a deeper S2-extensive subsite that facilitates stronger interactions with sitagliptin, whereas bacterial DPP-4 lacks this feature, reducing drug efficacy. Thus, the data reflect sitagliptin’s selective inhibition of host DPP-4 rather than bacterial suppression of host enzyme activity.

Comment 3: Figure 4a: It is a little misleading to attribute the lower DPP-4 activity upon fasting entirely to the decreased DPP-4 levels in BEVs, since host DPP-4 could also affect the readout.

We thank the reviewer for this important observation. We agree that it would be misleading to attribute the lower DPP-4 activity upon fasting entirely to decreased DPP-4 levels in BEVs without considering the potential contribution of host-derived DPP-4. Our proteomics analysis (Table S4a) confirmed that DPP-4 is present in host-derived EVs at comparable levels in both fasted and fed mice. Therefore, the observed reduction in total DPP-4 activity in the mixed EV population is most likely attributable to the decrease in bacterial DPP-4. This is now mentioned in the text, lines 382-385.

Comment 4: Figure 6: The modelling is interesting, but as mentioned in discussion, this needs to be verified by interaction and protease activity studies. Also, are the interactions functionally relevant if the bacterial serine proteases are significantly down-regulated already?

The structural modelling of serpin-protease interactions (e.g., serpin A3K/A1E with bacterial S9 proteases) is preliminary. While serpins form covalent complexes with proteases during inhibition, a process verifiable via SDS-PAGE due to migration shifts of SDS-stable complexes, functional validation through protease inhibition assays remains necessary (https://doi.org/10.1021/cr010170+ ) and this will certainly be investigated in the near future.

The observed reduction in bacterial serine proteases within BEVs during fasting may be explained by two complementary mechanisms. First, this downregulation likely reflects an adaptive response by Bt to nutrient scarcity, prioritising the expression of proteins involved in alternative energy acquisition such as the increased abundance of the oligosaccharide importer SusC over energetically costly protease production. Second, it is also possible that host-derived factors contribute to this regulation. For example, the increased presence of host serine protease inhibitors (serpins) and antimicrobial peptides like CRISC-2 α-defensin in host EVs during fasting could signal to the microbiota, either directly or indirectly, to suppress protease synthesis or alter EV cargo composition. Such host-microbe crosstalk may serve to further protect the gut epithelium from excessive proteolytic activity under conditions of nutrient stress, thereby maintaining intestinal homeostasis. This interpretation is now included in the manuscript, lines 395-416.

Comment 5: Line 386: ...BT_4393... should be BT_4193..?

Yes, this was corrected in the text.

Reviewer 2 Report

Comments and Suggestions for Authors

The authors did a TMT-based quantitative proteomics analysis of BEVs in the intestine of Bt germ-free mice with food or fasten treatment, aiming to detect the impact of nutrient stress on BEV protein composition. Serine protease family proteins were identified the differential expression significantly. 

The experiments are well designed and presented. I have a few clarifications and suggestions to improve this paper:

  1. LC-method and (DDA,  SPS-MS3) MS acquisition approach and database searching workflow using PD were missing, please add them.
  2. Please submit the MS raw files and original PD output into online public source, eg, PRIDE, or MassIVE.
  3. Please modify the volcano plot figures, some labels and dots were mixed.
  4. About Table1, it is not suitable to use FC to represent the decreased expression only, P-value is vital.
  5. About figure 3, why the resolution of a is lower than figure 3b; why the WT and GT showed the different result, please explain.
  6. In figure 4, why figure a did not show the label representing the significance. Please modify the P-value in figure (a), double chech with the supplementary table; revise the Protein ID in figure (b). 
  7. Please modify the tile, x-/y-axis in figure 5a, b.
  8. Please use the same decimal for fold change and p-value in Table 2.
  9. The Discussion is excessive long.

Author Response

Reviewer 2

We sincerely thank Reviewer 2 for their constructive feedback and positive assessment of our experimental design and presentation. Below, we address each clarification and suggestion to enhance the manuscript’s clarity and rigor.

Comment 1: LC-method and (DDA,  SPS-MS3) MS acquisition approach and database searching workflow using PD were missing, please add them.

Response: The LC-MS/MS methodology, including the DDA and SPS-MS3 acquisition approach and the Proteome Discoverer database searching workflow was missing and has now been added to the Materials and Methods section (lines 547-604).

Comment 2: Please submit the MS raw files and original PD output into online public source, eg, PRIDE, or MassIVE.

Response: Data Deposition: Raw MS files and PD output were uploaded to the PRIDE repository (Dataset ID: PXD062737). The Data Availability Statement was updated accordingly in the manuscript.

Comment 3: Please modify the volcano plot figures, some labels and dots were mixed.

Response: Figures 1d and 5b were redesigned with fewer annotations to reduce label overlap and make the plots clearer and easier to interpret.

Comment 4: About Table1, it is not suitable to use FC to represent the decreased expression only, P-value is vital.

Response: We have now introduced the P-value in Table 1 to provide a more comprehensive representation of statistical significance.

Comment 5: About figure 3, why the resolution of a is lower than figure 3b; why the WT and GT showed the different result, please explain.

Response: The lower resolution in Figure 3a resulted from compressed image export settings during panel assembly, which has been corrected. WT mice colonised with Bt BEVs contain the bacterial DPP-4 enzyme (BT_4193), which is only partially inhibited by sitagliptin, a drug that targets mammalian DPP-4. They also contain intestinal EVs that harbor host DPP-4 (Table S4a). In contrast, GF mice lack bacterial DPP-4, so any residual DPP-4 activity after sitagliptin treatment reflects only the mammalian (host) enzyme. This distinction is supported by both the experimental data and the literature, which confirm that BT_4193 is a functional bacterial DPP-4 homolog with different inhibitor sensitivity than the mammalian enzyme. This is described in lines 222-224 of the manuscript.

Comment 6: In figure 4, why figure a did not show the label representing the significance. Please modify the P-value in figure (a), double chech with the supplementary table; revise the Protein ID in figure (b). 

Response: The significance labels were missing in Figure 4a due to an oversight, but asterisks indicating statistical significance (P ≤ 0.001) were added and the P-value were double-checked against the supplementary table. In Figure 4b, the protein ID in the axis label was corrected to BT_2757 (asparaginase).

Comment 7: Please modify the tile, x-/y-axis in figure 5a, b.

Response: The title and axis labels in Figure 5a and 5b were updated for clarity. Figure 5b has now axes labeled "Log2(Fold Change)" and "-Log10(P-value).

Comment 8: Please use the same decimal for fold change and p-value in Table 2.

Response: The decimals for fold change and p-values have been standardized in both Table 1 and Table 2, with fold change rounded to two decimal places and p-values shown in scientific notation.

Comment 9: The Discussion is excessive long.

Response: The discussion was initially shortened by 25% through removing repetition and condensing paragraphs. However, in response to Reviewer 1’s comments, some content was reintroduced (lines 385–387 and 397–410), resulting in a net reduction of the discussion section by 16%. Based on these revisions and a careful evaluation of the content, we now feel confident that the discussion is well-balanced, clearly interprets the results, and provides appropriate context within the current literature.

Round 2

Reviewer 1 Report

Comments and Suggestions for Authors

The authors have addressed the comments.